# In Vitro Cytotoxic Activity of African Plants: A Review

**DOI:** 10.3390/molecules27154989

**Published:** 2022-08-05

**Authors:** Isabel Canga, Pedro Vita, Ana Isabel Oliveira, María Ángeles Castro, Cláudia Pinho

**Affiliations:** 1School of Health, Polytechnic Institute of Porto (ESS–P. Porto), 4200-072 Porto, Portugal; 2Department of Pharmaceutical Sciences, Pharmaceutical Chemistry Section, CIETUS, IBSAL, Pharmacy Faculty, University of Salamanca (USAL), 37007 Salamanca, Spain; 3Higher Polytechnic School Kwanza Norte, Kimpa Vita University, Uige 77, Angola; 4Research Centre in Health and Environment (CISA), School of Health, Polytechnic Institute of Porto (ESS–P. Porto), 4200-072 Porto, Portugal

**Keywords:** in vitro, cytotoxicity, African plants, cancer, cell lines

## Abstract

In African countries, cancer not only is a growing problem, but also a challenge because available funding and resources are limited. Therefore, African medicinal plants play a significant role in folk medicine and some of them are traditionally used for the treatment of cancer. The high mortality rate and adverse effects associated with cancer treatments have encouraged the search for novel plant-based drugs, thus, some African plants have been studied in recent years as a source of molecules with proven cytotoxicity. This review aims to discuss the cytotoxic activity, in vitro, of African plant crude extracts against cancer cell lines. For the period covered by this review (2017–2021) twenty-three articles were found and analyzed, which included a total of 105 plants, where the main cell lines used were those of breast cancer (MCF-7 and MDA-MBA-231) and colorectal cancer (HCT-116 and Caco-2), which are among the most prevalent cancers in Africa. In these studies, the plant crude extracts were obtained using different solvents, such as ethanol, methanol, or water, with variable results and IC_50_ values ranging from <20 µg/mL to >200 µg/mL. Water is the preferred solvent for most healers in African countries, however, in some studies, the aqueous extracts were the least potent. Apoptosis and the induction of cell cycle arrest may explain the cytotoxic activity seen in many of the plant extracts studied. Considering that the criteria of cytotoxicity activity for the crude extracts, as established by the American National Cancer Institute (NCI), is an IC_50_ < 30 μg/mL, we conclude that many extracts from the African flora could be a promising source of cytotoxic agents.

## 1. Introduction

Cancer is a generic term for a series of malignant diseases that is still a major health problem worldwide. The etiology of carcinogenesis involves distinct levels of regulation. In this process, normal cells acquire genetic and epigenetic changes that result in uncontrolled cell growth and, therefore, cancer. It is also known that reactive oxygen species (ROS) are involved in tumor formation through the activation of various oncogenic signaling pathways, DNA mutations, immune escape, the tumor microenvironment, metastasis, and angiogenesis [1]. Despite all the improvements in cancer therapy due to diagnostic and therapeutic progresses, cancer still has extremely high mortality rates [2]. Cancer is a global public health problem and the second leading cause of death in the United States [3]. Worldwide, an estimated 19.3 million new cancer cases with almost 10.0 million deaths occurred in 2020 [4].

One of the main problems with cancer cells is their ability to escape apoptosis due to unidentified mutations, resulting in cell accumulation, which consequently migrate to distinct parts of the body [5]. Thus, an effective anticancer drug should target cancer cells without affecting normal cells, which can be achieved by restoring the apoptosis machinery in the cancer ones [6] and by being able to overcome multidrug resistance (MDR). Cancer cells can rapidly acquire MDR, which can be associated with a variety of mechanisms, including the overexpression of adenosine triphosphate-binding cassette (ABC) efflux transporters [7], or the deletion/inactivation of important biomarkers of the tumorigenesis, such as tumor suppressor gene *p53* [8].

In Africa, cancer is reported as a critical public health problem [9], with increasing numbers related to aging, population growth, and an increase of risk factors, including smoking, obesity, and sexual and sedentary behaviors. In Africa, an estimated 1.1 million new cases and 711,429 deaths occurred due to neoplasms in 2020. By 2040, the burden of all neoplasms combined is forecasted to increase to 2.1 million new cases and 1.4 million deaths [10]. In addition, cancer is a challenge in African countries because, in general, available funding is limited, the lack of resources, and other major health problems [9].

Worldwide, more than 3000 plants have been reported to have anticancer properties [11]. Therefore, plants continue to be an important source of new cytotoxic agents due to the structural diversity of phytochemicals, and their use to combat multidrug resistance remains a challenge. The importance of African traditional plants in the prevention and treatment of diseases, including cancer, has already been shown [9]. However, despite the traditional use of medicinal plants, research on the cytotoxicity related to African plant extracts against cancer cells is scarce. Therefore, this review aims to gather the information about the cytotoxic activity, in vitro, of African plant crude extracts, describing mechanisms of action and the influence of extraction solvents.

To perform this review, the published literature from 2017 to 2021 on the African medicinal plants as sources of bioactive compounds with potential cytotoxic activity were collected from the online bibliographical databases: PubMed and ScienceDirect. To search for the in vitro cytotoxicity of African plants, the keywords cytotoxic, anticancer, antitumor, antiproliferative, cancer cell lines, and African plants were used in combination. This review included articles published in English that evaluated the cytotoxicity of the crude extracts by different assays, from plants collected/purchased fresh in African countries. The evaluations of the extract fractions or isolated compounds from medicinal plants were not included.

## 2. Results

A total of twenty-three reports, associated to the cytotoxic potential of 105 African medicinal plants, were retrieved from the databases selected. The in vitro studies from 2017 to December 2021, related to the cytotoxicity of the crude extracts from plants collected in Africa towards different human carcinoma cell lines, are collected in Table 1.

In Africa, many countries still depend on traditional plants for the management of several types of cancers due to the limited access to conventional medicine [12]. Despite progresses in chemotherapy for the treatment of cancer, other major problems appear (e.g., costs, side effects, and multidrug resistance) particularly in African countries [13].

Most of the plants came from South Africa (eight studies), followed by Cameroon (six studies), and then Morocco (two studies). Other countries such as Ghana, Ethiopia, Côte d’Ivoire, Egypt, Burkina Faso, Algeria, and Kenya were also included in the studies.

**Table 1 molecules-27-04989-t001:** In vitro cytotoxicity of extracts from medicinal plants collected from different countries in Africa against different cancer cell lines.

Country	Medicinal Plants	Extraction Solvents	Cancer Cell Lines	Results	Reference
South Africa	*Momordica balsamina* (leaves)	Acetone	Colorectal carcinoma (HT-29)	− Viability of HT-29 cells decreased in a concentration- and time-dependent manner. At concentrations > 50 µg/mL, a significant decrease in cell viability was seen (*p* ≤ 0.01). − *M. balsamina* suppressed cell invasion, cell adhesion and cell migration. − Cell invasion was associated to downregulation of NF-κB, TNF-α, NF-κB, MMP2, and MMP9, and to the upregulation of TIMP-3 proteins.	[14]
South Africa	*Sutherlandia frutescens* (leaves)	75% (V/V) Ethanol	Colon adenocarcinoma (DLD-1)	− Between concentrations of 22.2 µg/mL and 200 µg/mL, all the plant extracts induced cytotoxicity on DLD-1 cells in a dose-dependent manner. − Plants from Colesburg, Zastron, and Gansbaai 1 showed higher potency (IC_50_ values of 158.7, 172.7, and 176.7 µg/mL, respectively) compared to specimens from other localities in South Africa. − Plants from Colesburg had the highest anticancer activity (36.6% viability).	[15]
South Africa	*Tulbaghia violacea* (leaves)	Methanol Hexane Butanol	Cervical adenocarcinoma (HeLa and ME-180) Breast adenocarcinoma (MDA-MBA-231 and MCF-7)	− Most cytotoxic extract was the methanol one with the greatest effect on *p53* in all cancers − Concentration of 15 µM (butanolic extract) was considered optimal for treating cervical and breast cancer cell lines, based on the IC_50_ values in all the extracts. − *T. violacea* extracts inhibit cell proliferation in a cell line- and dose-dependent manner.	[16]
South Africa	*Opuntia stricta* (cladodes)	Water Acetone Ethanol	Human myeloid leukemia (U937)	− Only acetone extract showed mild cytotoxicity (IC_50_ = 110.1 µg/mL).	[17]
South Africa	*Cotyledon orbiculata* (leaves)	Water	Colorectal carcinoma (HCT116) Esophageal adenocarcinoma (OE33) Esophageal squamous carcinoma (KYSE70)	− *C. orbiculata* extract decreased the viability of KYSE70 (LC_50_ = 36.9 µg/mL), HCT116 (LC_50_ = 64.9 µg/mL), and OE33 cells (LC_50_ > 100 µg/mL) in a dose-dependent manner. − Apoptosis was induced by alternative splicing of hnRNPA2B1 and BCL2L1.	[13]
South Africa	*Asparagus laricinus* (cladodes) *Senecio asperulus* (roots)	Water Methanol Methanol:dichloromethane, 1:1 (V/V) Dichloromethane Hexane	Breast adenocarcinoma (MCF-7) Prostate adenocarcinoma (PC3)	− Methanol extract of *A. laricinus* showed cytotoxic effect towards MCF-7 cells (IC_50_ = 97.6 µg/mL), and almost no effect on non-cancerous Vero cells. − Dichloromethane extract of *S. asperulus* exhibited cytotoxic effect towards MCF-7 cells (IC_50_ = 69.15 µg/mL). − Methanol:dichloromethane extract of *A. laricinus* and hexane extract of *S. asperulus* were cytotoxic against breast and prostate cell lines studied.	[12]
South Africa	*Kedrostis africana* (tuber)	Water Ethanol	Cervical adenocarcinoma (HeLa)	− A significant decrease in the number of HeLa cells was not observed by aqueous extract of *K. africana*, at all tested concentrations (50–200 µg/mL). − For aqueous and ethanol extracts of *K. africana*, IC_50_ values are >200 µg/mL.	[18]
South Africa	*Centella asiatica* (leaves) *Curtisia dentata* (leaves) *Warburgia salutaris* (leaves)	Methanol Ethyl acetate Acetone Water	Breast adenocarcinoma (MCF-7) Cervical adenocarcinoma (HeLa) Colorectal adenocarcinoma (Caco-2) Lung adenocarcinoma (A549)	− Methanol and acetone extracts were more active than the ethyl acetate and aqueous extracts. − *C. asiatica*: acetone extract had the most significant activity (IC_50_ = 46.49 ± 0.04 μg/mL, A549 cell line) and aqueous extract was the least active (IC_50_ > 100 μg/mL for A549, Caco-2, and MCF-7 cell lines; and IC_50_ = 76.3 ± 0.06 μg/mL (HeLa)). − *C. dentata*: acetone extracts were the most cytotoxic (IC_50_ = 41.55, 45.13, 57.35 and 43.24 μg/mL against A549, HeLa, CaCo-2, and MCF-7 cell lines, respectively) (*p* ≤ 0.05). − *W. salutaris*: acetone extracts were the most cytotoxic (IC_50_ = 34.15 μg/mL for MCF-7).	[19]
Cameroon	*Tetrapleura tetraptera* (fruits)	Dichloromethane-methanol (1:1)	Resistant CEM/ADR5000 and sensitive CCRF-CEM leukemia cells; Colon cancer HCT116 (*p53+/+*) cells and t clone HCT116 (*p53−/−*); Glioblastoma U87MG cells and resistant U87MG.ΔEGFR cells; MDA-MB-231-pcDNA3 breast cancer cells and resistant subline MDA-MB-231-BCRP clone 23 cells; Hepatic carcinoma (HepG2)	− Crude extract of *T. tetraptera* showed cytotoxicity towards all the cancer cell lines (including drug-sensitive and -resistant phenotypes). − Crude extract of *T. tetraptera* showed cytotoxic effects with IC_50_ values ranging from 10.27 µg/mL, in CCRF-CEM leukemia cells, to 23.61 µg/mL, in colon cancer HCT116 (*p53−/−*) cells. − Apoptosis induced by crude extract of *T. tetraptera*, in CCRF-CEM cells, was MMP alteration-mediated and increased reactive oxygen species (ROS) generation.	[20]
Cameroon	*Fagara tessmannii* (bark)	Methanol	Resistant CEM/ADR5000 and sensitive CCRF-CEM leukemia cells; MDA-MB-231-pcDNA3 breast cancer cells and its resistant subline MDA-MB-231-BCRP clone 23 cells; Colon cancer HCT116 (*p53+/+*) cells and their t clone HCT116 (*p53−/−*) Glioblastoma U87MG cells and resistant subline U87MG.ΔEGFR cells; Hepatic carcinoma (HepG2)	− The effect of methanol extract of *F. tessmannii* in models of drug-resistant and drug-sensitive cell lines was studied. − IC_50_ values ranged from 17.34 µg/mL (towards U87MG.ΔEGFR glioblastoma cells) to 40.68 µg/mL (against CCRF-CEM leukemia cells) for crude extract. − Apoptosis induced by *F. tessmannii* bark extract in sensitive CCRF-CEM leukemia cells was mediated by increased ROS production.	[21]
Cameroon	*Ficus elastica* (wood of aerial roots) *Selaginella vogelli* (leaves)	Methanol	Cervical adenocarcinoma (HeLa)	− *S. vogelii* and *F. elastica* extracts had IC_50_ values at 20 µg/mL, for HeLa cell line. − Emetine exhibited an IC_50_ = 0.04 µM. − Extracts of both plants showed low cytotoxic effects on HeLa cell line.	[22]
Cameroon	*Sarcocephalus pobeguinii* (Roots, fruits, bark, and leaves)	Methanol (leaves/roots/bark) Dichloromethane/methanol (1:2) (fruits)	Breast adenocarcinoma (MCF-7) Cervical adenocarcinoma (HeLa) Colorectal adenocarcinoma (Caco-2) Lung adenocarcinoma (A549)	− Methanol extract from *S. pobeguinii* leaves had the highest cytotoxic activity. − Methanol extract from leaves is active against MCF-7 (IC_50_ = 26.94 μg/mL) and HeLa (IC_50_ = 10.19 μg/mL); methanol extract from bark is only efficient on HeLa cells (IC_50_ = 15.26 μg/mL). − Extract from the fruits was significantly more toxic to non-cancerous (Vero) cells (LC_50_ = 601.42 µg/mL) (*p* < 0.05) than Caco-2 cell line (IC_50_ = 721.03 µg/mL). − Methanol extract (roots) showed the higher IC_50_ values for MCF-7 and Caco-2.	[23]
Cameroon	*Moringa oleifera* (leaves and seeds)	Water	Human lymphoid (Jurkat E6-1) Human leukemia monocytic (THP1)	− Aqueous extract from leaves and seeds had anti-proliferative and pro-apoptotic effects only on cancer cells (not on peripheral blood mononuclear cells—PBMCs). − Pro-apoptotic effect seen with aqueous extract is related with decreased BCL2 levels and sirtuin-1 (SIRT1) protein expression.	[24]
Cameroon	*Ananas comosus* (peels) *Arachis hypogaea* (leaves and twigs) *Artocarpus heterophyllus* (leaves) *Camelia sinensis* (leaves) *Citrus sinensis* (fruits) *Cola pachycarpa* (leaves) *Coula edulis* (fruits) *Curcubita pepo* (pericarp) *Curcuma longa* (rhizomes) *Lycopersicon esculentum* (twigs and leaves) *Mangifera indica* (leaves and bark) *Myristica fragrans* (seeds) *Persea Americana* (bark) *Physalis peruviana* (twigs) *Psidium guajava* (bark) *Raphia hookeri* (fruits) *Rubus fellatae* (leaves) *Tristemma hirtum* (leaves)	Methanol	Resistant CEM/ADR5000 and sensitive CCRF-CEM leukemia cells; MDA-MB-231-pcDNA3 breast cancer cells and their transfectant subline MDA-MB-231-BCRP clone 23; Colon cancer HCT116 (*p53+/+*) cells and their knockout clone HCT116 (*p53−/−*); Glioblastoma U87MG cells and their resistant subline U87MG.ΔEGFR	− Doxorubicin (positive control) and 13 out of 21 plant extracts had IC_50_ values < 80 µg/mL, against the sensitive leukemia CCRF-CEM cells. − Six out of the 13 active extracts showed IC_50_ values below 30 µg/mL. − *C. longa* methanol extract: IC_50_ = 6.25 μg/mL (HCT116 *p53−/−*) and IC_50_ = 10.29 μg/mL (MDA-MB-231-BCRP cells). − *L. esculentum* methanol extract: IC_50_ = of 9.64 μg/mL (MDA-MB-231 cells) and IC_50_ = 57.74 μg/mL (HepG2 cells). − *P. guajava*: IC_50_ value = 1.29 μg/mL (CEM/ADR5000 cells and IC_50_ = 62.64 μg/mL (MDA-MB-231 cells). − Each of the six plant extracts showed a degree of resistance (DR) < 1.00 in at least one type of malignant cells. The DR of all extracts were lower than that of doxorubicin in all cancer cells evaluated.	[25]
Morocco	*Calendula arvensis* (flowers)	Hexane Methanol Water	Human cancer of myeloid cells	− Methanol extract was the most cytotoxic (IC_50_ value = 31 µg/mL), with maximum 89% inhibition at the concentration 100 mg/mL at 24 h (*p* < 0.05). − Methanol and aqueous extracts (flowers) had promising antimyeloid cancer efficacy.	[26]
Morocco	*Ormenis eriolepis* (aerial parts)	n-Hexane Methanol	T lymphocyte cell line (Jurkat) Mantle cell lymphoma (Jeko-1) Glioblastoma (LN229) Prostate adenocarcinoma (PC-3)	− Hexanic extract showed high effect against Jurkat, Jeko-1, LN229, and PC-3 cells, but not against normal cell lines. − Hexanic extract showed IC_50_ values of 19.31 µg/mL (PC-3) and 41.67 µg/mL (LN229) and induced G1 (in Jurkat, Jeko-1, and LN22 cell lines) and G2/M (in PC-3 cell line) phases’ cycle arrest.	[27]
Ghana	*Aframomum melegueta* (seeds, roots/rhizome) *Alstonia boonei* (leaves, roots) *Baphia nitida* (leaves) *Desmodium adscendens* (leaves, stems) *Ficus asperifolia* (leaves, stem bark) *Mansonia altissima* (stem bark) *Paullinia pinnata* (stem) *Spathodea campanulate* (leaves, stem bark) *Terminalia superba* (leaves, stem bark, roots) *Triplochiton scleroxylon* (leaves, stem bark)	Ethanol–water (1:1)	Hepatic carcinoma (HepG2) Breast adenocarcinoma (MDA-MB-231 and MCF-7) Epidermoid carcinoma (A431) Prostate adenocarcinoma (LNCaP) Lung adenocarcinoma (A549) Gastric adenocarcinoma (AGS) Leukemia (HL-60 and REH) Ewing’s sarcoma (CADO-ES1 and RDES)	− From all the plant extracts, only two decreased cell viability in cancer cell lines evaluated. − *A. boonei* extract (leaves) and *P. pinnata* extract (stems) showed IC_50_ values of about 50 µg/mL (IC_50_ = 42.7, 47.5, and 50.9 for leaves of *A. boonei* in A549, MCF-7, and LNCap cell lines, respectively; IC_50_ = 42.8, 43.1, 47.2, and 47.6 for stems of *P. pinnata* in HepG2, MCF-7, LNCap, and AGS-7 cell lines, respectively). − Preliminary TLC investigations showed oligomeric and polymeric proanthocyanidins as the predominant class of phytochemicals in the *P. pinnata* hydroethanolic extract. − The presence of 15-hydroxyangustilobine A (vallesamine-type indole alkaloid) was seen in the bioassay-guided fractionation of the *A. boonei* extract and considered as the active principle responsible for the cell cycle arrest of MCF-7 cells at G2/M phase (MCF-7 cells), triggering cells at least partially into apoptosis.	[28]
Ethiopia	*Acmella caulirhiza* (leaves) *Acokanthera schimperi* (leaves) *Ajuga leucantha* (leaves) *Aloe debrana* (roots) *Cineraria abyssinica* (leaves) *Clausena anisate* (leaves) *Clematis simensis* (leaves) *Cleome brachycarpa* (leaves) *Croton macrostachyus* (bark) *Dorstenia barnimiana* (roots) *Euphorbia schimperiana* (roots) *Gnidia involucrate* (roots) *Hydrocotyle mannii* (leaves) *Kalanchoe petitiana* (leaves) *Kniphofia foliosa* (roots) *Leonotis ocymifolia* (leaves) *Pentarrhinum insipidum* (roots) *Rumex nervosus* (roots) *Salvia nilotica* (whole plant) *Sida schimperiana* (roots and leaves) *Thymus schimperi* (leaves) *Vernonia auriculifera* (leaves)	80% Methanol	Breast adenocarcinoma (MCF-7) Lung carcinoma (A427) Urinary bladder carcinoma (RT-4) Cervical adenocarcinoma (SiSo) Large cell lung carcinoma (LCLC-103H) Pancreatic carcinoma (DAN-G) Ovarian carcinoma (A2780) Esophageal squamous carcinoma (KYSE-70) Acute myeloid leukemia (HL-60) Human myeloid leukemia (U-937)	− A first screening was performed where extracts were tested at a concentration of 50 µg/mL towards four cancer cell lines (A427, MCF-7, RT-4, and SiSo). − Four out of 22 plant extracts (*A. schimperi, E. schimperiana, K. foliosa, and K. petitiana*) showed relevant cytotoxic activity and were selected for secondary screening against all the adherent and suspension cell lines. − *A. schimperi* showed potent cytotoxic activity towards all cell lines studied (IC_50_ values ranging from 1.87 to 10.31 µg/mL). − *E. schimperiana* showed potent cytotoxic activity against A427, SiSo, and RT-4 cell lines at concentrations ranging from 1.85 to 3.28 µg/mL. − IC_50_ values presented by *K. petitiana* extracts ranged from 2.09 to 10.41 µg/mL. − *K. foliosa* showed anti-proliferative effect in all cell lines, with IC_50_ values ranging from 14.54 to 27.06 µg/mL. − *E. schimperiana*, *A. schimperi*, *K. foliosa*, and *K. petitiana* extracts demonstrated selective cytotoxicity against suspension cell lines (HL-60 and U-937) compared to Peripheral Blood Mononuclear Cell (PBMC).	[29]
Côte d’Ivoire	*Bridelia ferruginea* (leaves and stem barks)	Methanol Ethyl acetate Water	Colorectal carcinoma (HCT116)	− A lethality assay was conducted in *Artemia salina* to study the cytotoxicity of the *B. ferruginea* extract. LC50 value from brine shrimp assay was below 2 mg/mL. − Inhibition of HCT116 cell viability was seen in a concentration-dependent manner. − Inhibition of HCT116 cell viability induced by stem bark methanol extract could be related to its rich phenolic content (e.g., catechin fraction).	[30]
Egypt	*Brassica nigra* (seeds)	50% (V/V) Ethanol	Human non-small cell lung carcinoma (A549 and H1299)	− *B. nigra* extract showed cytotoxic activity against A549 (IC_50_ = 32.02 µg/mL) and H1299 cell lines (IC_50_ = 25.38 µg/mL). − Apoptosis induced by *B. nigra* in a time- and concentration-dependent manner related to increased caspase-3 activity. − Treatment of A549 and H1299 cell lines with *B. nigra* extract result in significant S and G2/M phases’ arrest of cell cycle (*p* < 0.01 and *p* < 0.001). − Suppression of migratory and invasive properties of A549 and H1299 by *B. nigra* extract. − *B. nigra* extract downregulated the expression of matrix metalloproteinases (MMP2 and MMP9) and Snail, and upregulated expression of E-cadherin at mRNA and protein levels.	[31]
Burkina Faso	*Lantana ukambensis* (whole plant)	Dichloromethane	Colorectal carcinoma (HCT-116 and HT-29)	− Significant cytotoxic effect towards HCT-116 (20 µg/mL) and HT-29 (80 µg/mL) cell lines was observed with *L. ukambensis* crude extract after 48 h of incubation (*p* < 0.0001). − IC_50_ values = 23.05 µg/mL and 106.81 µg/mL for HCT-116 and HT-29, respectively.	[32]
Algeria	*Heliotropium bacciferum* (aerial parts)	Chloroform Methanol	Colorectal carcinoma (HCT116) Colorectal adenocarcinoma (DLD1)	− Evaluation of antiproliferative activity of *H. bacciferum* extracts (concentrations ranging from 1.000 to 0.025 mg/mL). − *H. bacciferum* chloroform extract showed an inhibitory effect on the growth of DLD1 (IC_50_ = 62 µg/mL) and HCT116 (IC_50_ = 95 µg/mL) cell lines in a concentration-dependent manner. − *H. bacciferum* methanol extract did not show any cytotoxic effect.	[33]
Kenya	*Abrus precatorius* *Aeschynomene abyssinica* *Albizia gumífera*, *Aloe volkensii* *Bridelia micrantha* *Conyza sumatrensis* *Croton macrostachyus* *Cyphostemma serpens* *Entada abyssinica*, *Ficus thonningii* *Fuerstia africana*, *Futumia africana* *Harungana madagascariensis* *Ipomoea cairica* *Microglossa pyrifolia* *Momordica foetida* *Moringa oleífera* *Ocimum gratissimum*, *Olea hotch* *Phyllanthus sapialis*, *P. fischeri* *Prunus africana* *Psydrax schimperiana* *Rotheca myricoides* *Senna didymobotyra* *Shirakiopsis elliptica*, *Sida rhombifolia* *Spathodea campanulate* *Synsepalum cerasiferum* *Tragia brevipes*, *Trichilia emetica* *Triumfetta rhomboidei*, *Vernonia lasiopus* *Zanthoxylum rubescens*, *Z. gilletii*	Dichloromethane/methanol (organic) and water	Sensitive and drug-resistant human cancer cell lines: Sensitive CCRF-CEM and multidrug-resistant P glycoprotein-overexpressing CEM/ADR5000; Wild-type HCT116 (*p53+/+*) and knockout HCT116 (*p53−/−*) colon cancer cells; Breast cancer cells transduced with control vector (MDAMB-231-pcDNA3) or with cDNA for the breast cancer resistance protein BCRP (MDA-MB-231-BCRP clone 23) Wild-type U87MG cells and U87MG glioblastoma multiforme cells transfected with an expression vector harboring an epidermal growth factor receptor (EGFR) gene with a genomic deletion of exons 2 through 7 (U87MG.ΔEGFR)	− Screening results: initially, 34 organic and 19 aqueous extracts tested. − Drug-sensitive CCRF-CEM and multidrug-resistant CEM/ADR5000 cells were inhibited by organic extracts (*H. madagascariensis* and *P. africana*) by more than 80%. − Some organic extracts were more cytotoxic to multidrug-resistant CEM/ADR5000 cells than to sensitive CCRF-CEM cells. − MDA-MB-231 cells exerted collateral sensitivity towards 4 organic extracts (*H. madagascariensis*, *Z. rubescens*, *B. micrantha*, *Z. gilletii*). − Organic extract of *H. madagascariensis* inhibited both wild-type and knockout cell lines by more than 80%. *B. micrantha* and *H. madagascariensis* organic extracts exerted the strongest cytotoxicity towards both U87.MG cell lines. − *P. africana* organic extracts showed the best cytotoxic activity, inhibiting the proliferation of 7 out of 8 tested cancer cell lines (IC_50_ < 40 µg/mL). − Combination treatments: some extracts exhibited enhanced cytotoxicity towards cancer cells, if applied in combination with other extracts.	[34]

## 3. Discussion

### 3.1. Cytotoxicity

Countries in Africa are facing an increase in the incidence of cancer. The most prevalent in African females is breast cancer (27.7%), followed by cervical cancer (19.6%) [35]. In South Africa, colorectal cancer occupies the third place for women [36]. Meanwhile, prostate (18.1%), followed by liver (9.7%) and colorectal cancers (6.9%) were the most prevalent among African males [35]. According to the World Health Organization (WHO), it is estimated that there will be an increase in the incidence and mortality rates of other types of cancer for the next two decades [37]. Regarding cell lines used in the analyzed studies, the most selected for the evaluation of cytotoxicity were breast cancer (MCF7 and MDA-MBA-231) and colorectal cancer (HCT-116 or Caco-2) because of the high and increased incidence rates of these cancers.

Different assays were performed to evaluate the cytotoxicity effects of the studied African plant extracts, including a crystal violet cell antiproliferation assay [29], resazurin reduction assay [20,21,22,25,34], or tetrazolium-based colorimetric cell proliferation assays such as MTT [14,16,17,19,23,26,28,29,30,31,33], MTS [32] and WST-1 [13]. There were studies where cell numbers were determined using Hoechst 33342/Propidium Iodide (PI) staining [12,18], trypan blue [24], or were based on the quantification of Adenosin Triphosphate (ATP), signaling the presence of metabolically active cells [15,27].

According to the American National Cancer Institute USA (NCI), the criteria of cytotoxicity activity for botanicals/crude extracts is IC_50_ < 20 µg/mL or 10 µM upon 48 h or 72 h incubation [38]. The NCI considers an IC_50_ upper limit criteria of 30 µg/mL as a promising crude extract for purification [39]. However, other authors such as Ayoub et al. (2014) [40] consider that higher values and a plant extract could be effective as being cytotoxic at concentrations up to 100 μg/mL. According to results showed in Table 1, some extracts were active below 100 μg/mL, therefore, they can be considered promising sources for the development of novel anticancer drugs.

Regarding plants collected in South Africa, some extracts showed IC_50_ values lower than 100 µg/mL. For example, in the study of Makhafola et al. (2020) [13], the crude extract of *C. orbiculata* decreased cell viability in a dose-dependent manner of HCT116 and KYSE70 cell lines with LC_50_ values of 64.9 and 36.9 µg/mL, respectively, showing that the KYSE670 esophageal cancer cell line was most susceptible to the extract [13]. In addition, Mfengwana et al. (2019) [12] showed that *A. laricinus* methanol extract (IC_50_ value of 97.6 µg/mL) and *S. asperulus* dichloromethane extract (IC_50_ value of 69.15 µg/mL) showed cytotoxic activity against MCF-7 cancer cells [12]. Finally, acetone extract from *C. dentata* [19] significantly (*p* ≤ 0.05) revealed IC_50_ values of 41.55, 45.13, 57.35, and 43.24 µg/mL against A549, HeLa, CaCo-2, and MCF-7 cell lines, respectively. The acetone extracts from *W. salutaris* showed an IC_50_ value of 34.15 µg/mL against the MCF-7 cell line [19]. However, some plant crude extracts from South Africa revealed higher IC_50_ values. The studies performed by Serala et al. (2021) [14] and Motadi et al. (2020) [16] with *M. balsamica* and *T. violacea* extracts, respectively, do not determine the exact IC_50_, but indicate that the extracts affect the viability of the tumoral cell lines in a dose-dependent manner up to the highest concentrations tested (50 and 20 µg/mL, respectively). Other studies revealed IC_50_ values higher than 100 µg/mL. For example, in the study of Zoyane et al. (2020) [15] with *S. frutescens* extracts produced from plants growing at different geographic localities in South Africa, IC_50_ values confirmed the relatively higher potency of plants from Colesburg, Zastron, and Gansbaai 1 (158.7, 172.7, and 176.7 µg/mL, respectively) [15]. Furthermore, in the study of Izuegbuna et al. (2019) [17], the acetone-dried extract of *O. stricta* showed mild cytotoxicity (IC_50_ = 110.1 µg/mL), and Unuofin et al. (2018) [18] showed IC_50_ values for aqueous and ethanol extracts of *K. africana* > 200 µg/mL.

For plants collected in Cameroon, IC_50_ values below or around 30 µg/mL were shown in nearly all the studies analyzed. Mbaveng et al. (2021) [20] evaluated the cytotoxicity of the fruit’s crude extract obtained from *T. tetraptera* on different cancer cell lines. The crude extract displayed IC_50_ values below 20 µg/mL on seven out of nine cancer cell lines tested. The IC_50_ values obtained varied from 10.27 µg/mL (in CCRF-CEM leukemia cells) to 23.61 µg/mL (against HCT116 *p53−/−* cancer colon cells) [20]. Mbaveng et al. (2019) [21] also determined the cytotoxicity of the *F. tessmannii* methanol bark extract, after 72 h incubation, in seven cancer cell lines. IC_50_ values below 20 µg/mL were recorded with the same crude extract in three cancer cell lines (MDA-MB-231-pcDNA, IC_50_ = 19.43 ± 0.88 µg/mL; MDA-MB-231-BCRP, IC_50_ = 18.87 ± 1.16 µg/mL; U87MG.ΔEGFR, IC_50_ = 17.34 ± 1.37 µg/mL) [21]. In a previous study of Mbaveng et al. (2018) [25], *C. longa* rhizomes and *L. esculentum* leaves displayed IC_50_ values below 20 µg/mL in most cancer cell lines evaluated. Additionally, Mbosso Teinkela et al. (2018) [22] showed IC_50_ values of 20 µg/mL for the methanol extract of *S. vogelii* leaves and the wood methanol extract of *F. elastica* aerial roots [22]. Mfotie Njoya et al. (2017) [23] also found IC_50_ values below 30 µg/mL on MCF-7 and HeLa cells for the *S. pobeguinii* methanol extracts from leaves and bark. Finally, Potestà et al. (2019) [24] studied the antiproliferative activity of boiled and frozen aqueous extracts from *M. oleifera*, a well-known species used for medicinal and nutritional purposes. Both preparations, boiled and frozen, showed EC_50_ values ranging between 10 and 20 µg/mL for THP1 leukemia cells. For Jurkat cells, the boiled preparations had EC_50_ values over 100 µg/mL, although the EC_50_ for the frozen extracts ranged between 1 and 11 µg/mL, showing an interesting antiproliferative and cytotoxic effect [24].

In the two studies performed with plants from Morocco, a methanol extract of *C. arvensis* (flowers) was the most significant anti-myeloid cancer agent, showing an IC_50_ value of 31 µg/mL [26]. Similar IC_50_ values were showed in the study of Belayachi et al. (2017) for PC-3 (19.31 ± 4.88 µg/mL) and for LN229 cells (41.67 ± 1.98 µg/mL) upon treatment with the hexane extract of *O. eriolepis* [27].

For plants collected in other African countries (e.g., Ethiopia, Egypt, Burkina Faso, and Kenya), IC_50_ values up to 30 µg/mL were also seen. Tesfaye et al. (2021) [29] showed the potent activity of two plants out of 22 collected in Ethiopia against all 10 cell lines evaluated, namely *A. schimperi* extract (80% methanol), with IC_50_ values from 1.87 ± 0.4 to 10.31 ± 3.45 µg/mL, and *K. petitiana* extracts which had IC_50_ values from 2.09 ± 0.43 to 10.41 ± 5.59 µg/mL [29]. Moreover, *B. nigra* extract, collected from Egypt and evaluated in the study of Ahmed et al. (2020) [31], showed IC_50_ values of 32.02 and 25.38 µg/mL against the A549 and H1299 cell lines, respectively. Sawadogo et al. (2020) [32] showed for the *L. ukambensis* crude extract (plant collected from Burkina Faso) an IC_50_ value of 23.05 ± 1.56 µg/mL in the HCT-116 cell line. Finally, in the study of Ochwang’I et al. (2018) performed with 35 medicinal plants from Kenya, the highest concentration tested (40 µg/mL) showed an IC_50_ value of 30 µg/mL [34]. For example, the *P. africana* organic extracts showed the best cytotoxic activity, inhibiting the proliferation in seven out of eight tested cancer cell lines (IC_50_ < 40 µg/mL). However, higher IC_50_ values were also seen in other studies. For example, Spiegler et al. (2021) [28] evaluated the hydroethanolic extracts from ten plants collected in Ghana and only the leaf extract from *A. boonei* and the stem extract of *P. pinnata* showed IC_50_ values around 50 µg/mL in some of the cancer cells tested. Mahomoodally et al. (2021) [30] studied the pharmacological potential of *B. ferruginea*, collected from Côte d’Ivoire and found an antiproliferative effect against the HCT116 cell line in a concentration-dependent manner up to 200 µg/mL [30]. Aïssaoui et al. (2019) [33] described the cytotoxic effects in cancer cell lines of extracts from *H. bacciferum*, a plant collected in Algeria. In this study, *H. bacciferum* chloroform extract showed a concentration-dependent inhibitory effect on the growth of the treated cancer cell lines with IC_50_ values of 95 µg/mL on HCT116 and 62 µg/mL on DLD1 [33].

An ideal anticancer agent should have more of a cytotoxic effect on cancer cell lines than in noncancer cells. Therefore, high selectivity towards cancer cells is a desired property for these agents. In some plants referred in Table 1, this preliminary selectivity was observed. In the study of Mfengwana et al. (2019) with two South African plants, the dose-dependent cytotoxicity effect of the methanol extract of *A. laricinus* and the dichloromethane extract of *S. asperulus* were shown to be selective to the MCF-3 and PC3 cell lines, with a negligible effect on Vero cells [12].

Regarding the results from plants collected in Cameroon, Potestà et al. (2019) showed that particularly boiled *M. oleifera* extract showed a specific anti-proliferative activity on cancer cells, but not on the Peripheral Blood Mononuclear Cell (PBMC) [24]. In addition, the results of Mfotie Njoya et al. (2017) demonstrated that the methanol extract from leaves of *S. pobeguinii* was selectively cytotoxic to cancer cell lines compared to the normal Vero cells, with the Selectivity Index (SI) ranging from 3.15 to 18.28 on the four cancer cells lines (MCF-7, HeLa, Caco-2, and A549), suggesting the potential and antiproliferative effect of this extract [23]. Furthermore, in the case of some plants collected from nine districts in Ethiopia, Tesfaye et al. (2021) demonstrated that *E. schimperiana*, *A. schimperi*, *K. foliosa*, and *K. petitiana* extracts (80% methanol) had selective cytotoxicity against suspension cell lines (U-937 and HL-60) when compared to their effect on PBMC. The IC_50_ value of all extracts against PBMC was ˃50 µg/mL [29].

### 3.2. Mechanisms of Action

Anticancer effects of plants are related to the suppression of cancer-stimulating enzymes, repairing DNA, stimulating the production of antitumor enzymes in cells, increasing body immunity, and inducing antioxidant effects [41]. Novel therapeutic strategies against cancer will mediate the induction of the apoptosis pathway or cell cycle arrest. Caspases activation, an increase in ROS production, and MMP disruption have been involved in the induction of apoptosis of several botanicals from African flora [16]. Apoptosis is the process of programmed cell death, triggered by physiological and pathological stimuli, in which a cell dies as part of its normal process of development, when the immune system has ordered it to die, or due to a lack of growth factors. However, cancer cells can evade apoptosis, continuing to proliferate and even become resistant to chemotherapy [12]. Apoptosis has become the most investigated mode of action of cytotoxic drugs and involves an energy-dependent cascade of molecular events. For example, caspases activation and the loss of mitochondrial membrane potential are events involved in apoptosis [21]. In this section, the mechanisms of action described in the previous African plant studies are discussed.

Regarding the plants collected in South Africa, Motadi et al. (2020) showed the involvement of *T. violacea* hexane extracts in the induction of apoptosis, caused by the activation of *p53*, which can be related to the solvent used in the extraction procedure [16]. In some studies, caspase activity has been shown to be activated in methanol extracts [42,43]. However, Motadi et al. (2020) observed an increased caspase-3/7 activity, especially in cervical cancer cells treated with the hexane extract of *T. violacea* [16]. Moreover, Makhafola et al. (2020) concluded that *C. orbiculata* aqueous extract had an anti-proliferative effect in HCT116, KYSE70, and OE33 cancer cells, mediated by apoptosis induced by the alternative splicing of hnRNPA2B1 (an RNA-binding protein) and BCL2L1 (an important apoptosis-regulating gene) [13]. It is also known that hnRNPA2B1 plays a significant role in cancer progression, acting as an oncogene in the development of certain types of cancers [44]. Finally, Mfengwana et al. (2019) observed that *A. laricinus* methanol extract shows cytotoxic effects in MCF-7 cancer cells due to apoptosis when compared with a negative control (medium only) and positive control (melphalan). The authors also found that *S. asperulus* dichloromethane extracts showed cytotoxicity against prostate PC3 cancer cells (IC_50_ values of 69.25 µg/mL) due to cell cycle arrest at the G2 and early mitotic (G2/M) phases [12].

In the studies performed with plants from Cameroon, Mbaveng et al. (2021) showed that doxorubicin induced S and G2/M phase cycle arrest in CCRF-CEM cells, whilst crude extract of *T. tetraptera* fruits induced it in the G0/G1 phase. The crude extract induced apoptosis in those cells through matrix metalloproteinases (MMP) alteration and by increasing ROS production [20]. In another study, Mbaveng et al. (2019) demonstrated that *F. tessmannii* methanol bark extract induced apoptosis in CCRF-CEM cells [21]. Additionally, in a previous study, Mbaveng et al. (2018) showed that *C. longa* rhizomes, *P. guajava* bark, and *L. esculentum* leaves induced apoptosis via caspase activation and increased ROS generation in CCRF-CEM cells. Additionally, *C. longa* rhizomes and *P. guajava* bark induced mitochondrial membrane potential depletion, which contributed to apoptosis induction too [25].

The bioassay-guided fractionation of the *A. boonei* extract, from Ghana, performed by Spiegler et al. (2021), revealed the presence of an alkaloid (15-hydroxyangustilobine A), which, at concentrations ≥ 60 µM, caused an increase in the number of cells in the G2/M phase, which was higher than cells in apoptosis at the same concentration [28]. Moreover, caspase-3 kinetic activity increased in A549 and H1299 cancer cell lines treated with *B. nigra* extract in a time-dependent manner, suggesting that the plant extract collected in Egypt has the ability to induce apoptosis [31]. Ahmed et al. (2020) also showed that *B. nigra* ethanolic extract significantly delayed and arrested, at the S and G2/M transition, both A549 and H1299 cells in a concentration-dependent manner [31].

Cancer metastasis is a multi-cascade process involving distinct stages: cell migration, invasion, attachment, and angiogenesis [45]. Extracellular matrix degradation is considered the most crucial step of the metastatic process, being facilitated by MMPs [46]. Their expression can also be stimulated by proinflammatory cytokines such as tumor necrosis factor-alpha (TNF-α), interleukins (IL)-1β, and IL-6 [47]. In consequence, inhibiting the expression and/or activity of MMP2 and 9 could represent an essential role in the inhibition of cancer metastasis. Serala et al. (2021) showed that *M. balsamina* (a plant collected in South Africa) downregulated, in a significant manner, the expression of MMP2 and MMP9, confirming, therefore, their role in the HT-29 cell invasiveness reduction. *M. balsamina* also downregulated the expression of TNF-α and NF-κB proteins, which suggests that the *M. balsamina* extract might inhibit MMP2 and MMP9 expression at a transcriptional level in addition to inhibiting MMP9 activity through tissue inhibitors, namely TIMP-3 [14]. Furthermore, the potential migration and invasion inhibitions by the ethanolic extract of the Egyptian *B. nigra* was also evidenced by the downregulation of MMP2, MMP9, and Snail genes, and the upregulation of the E-cadherin gene (hallmark of cancer metastasis) [31].

Several African medicinal plants have been reported as having antiproliferative properties against MDR cancer cells, which is interesting since novel drugs with activity against tumors that do not respond to established anticancer drugs are urgently needed. Therefore, Ochwang’I et al. (2018) [34] analyzed the ability of plant extracts towards drug resistance mediated by tumor suppressor TP53 functional loss or by the mutational activation of the EGFR oncogene. Results showed collateral sensitivity, where some organic (*F. africana*, *S. cerasiferum*, *M. pyrifolia*, and *B. micrantha*) and aqueous extracts (*H. madagascariensis*, and *Z. gilletii*) presented higher cytotoxicity to multidrug-resistant CEM/ADR5000 cells than to sensitive CCRF-CEM cells [34]. Mbaveng et al. (2018) have shown that some Cameroon extracts studied (*C. pachycarpa*, *C. longa*, *L. esculentum*, *P. guajava*, *P. americana*, and *P. peruviana*) could be used against MDR cancer cell lines [25]. A recent study from Mbaveng et al. (2021) [20] continued to use various models of resistant cancer cells. The hypersensitivity of MDA-MB-231-BCRP and U87MG.ΔEGFR cells compared to their sensitive congeners (MDA-MB-231-pcDNA and U87MG) was observed for the crude extract of *T. tetraptera* (degree of resistance of 0.28 and 0.63, respectively) [20].

### 3.3. Effect of the Extract Solvents

It is recognized that the choice of the solvent used in an extraction defines the extract’s chemical profile and, potentially, influences its cytotoxic effects. In fact, different solvents were used in the studies, especially methanol, water, ethanol, and hexane, with mixed results. Water, taking into consideration its availability, appears as the preferred solvent used by most traditional healers. However, water only extracts polar bioactive compounds. In some studies, performed with South African plants, Soyingbe et al. (2018) [19] showed that aqueous extracts of *C. asiatica*, *W. salutaris*, and *C. dentata* (leaves) were the least active, with IC_50_ values > 100 µg/mL (A549, Caco-2 and MCF-7) and 76.3 ± 0.06 µg/mL (HeLa) [19]. Comparable results were seen by Unuofin et al. (2018), where *K. africana* aqueous extract presented no significant decrease in cell number in all the studied concentrations (50–200 µg/mL) [18]. For that reason, many authors worked with other solvents to increase the extraction of compounds of varying polarities.

Solvents less predominant in the studies such as ethyl acetate and 1-butanol may have polar molecules, including saponins and polyphenols such as tannins, anthocyanins, phenolic acids, flavonoids, and stilbenes, which are recognized as having preventive and curative anticancer benefits [32]. Methanol is also an example of a solvent used for most of the authors in Table 1. In some cases, methanol was the most active, as compared to other extracts such as ethyl acetate or aqueous extracts. However, we must consider the plant extract used. For example, in their study, Mbaveng et al. (2021) concluded that a suitable extraction solvent, such as a dichloromethane-methanol (1:1) mixture should be considered if using the fruits of *T. tetraptera* as a cytotoxic agent [20]. Mahomoodally et al. (2021) suggested the polar extracts from *Bridelia* species as good candidates for future studies aiming to explore in vivo anticancer activity [30]. Motadi et al. (2020) showed that the methanol extract of *T. violacea* (leaves) showed the highest cytotoxicity of the extracts [16]. However, the opposite was observed by Aïssaoui et al. (2019) who demonstrated that *H. bacciferum* methanol extract did not show any cytotoxic effects [33].

The diverse biological properties of plant extracts can also be due to its chemical composition variability. Regarding the studies performed with plants from South Africa, for Zonyane et al. (2020) [15], it is clear that anticancer activity shown for extracts of *S. frutescensare* is not necessarily strongly correlated to the flavonoids. Some other important chemicals found in that extract include sutherlandins isomers, triterpenoids, and cycloartenol glycosides (sutherlandiosides) [15]. In the study performed by Izuegbuna et al. (2019), polyphenols (phenols, flavonoids, flavonols, proanthocyanidins, tannins) were the major compounds found in the several *O. stricta* extracts. However, the non-cytotoxic effect of some extracts of *O. stricta* cladodes against U937 cells may be explained by an insufficient amount of compounds [17]. Finally, Soyingbe et al. (2018) showed that *C. asiatica* acetone extract had the most significant activity (IC_50_ = 46.49 ± 0.04 μg/mL) for A549 cells [19]. In a study by Naidoo et al. (2017), leukemic THP-1 cells’ viability was not significantly altered by *C. asiatica* ethanolic leaf extract (0.2–0.8 mg/mL) as compared to the control [48].

Analyzing the studies related to plants collected in Cameroon, Mbaveng et al. (2021) [20] showed that the active compounds of the dichloromethane-methanol (1:1) extract of *T. tetraptera* include betulinic acid, naringenin, luteolin, 3-*O*-[6′-*O*-undecanoyl-*β*-d-glucopyranosyl] stigmasterol, olean-12-en-3-*β*-*O*-d-glucopyranoside, 3-*O*-*β*-d-glucopyranosyl-(1→6)-*β-*d-glucopyranosylurs-12-en-28-oic acid, and 3-*O*-*β*-d-glucopyranosyl-(1→3)-*β*-d-glucopyranosyl-27-hydroxyolean-12-en-28-oic acid [20]. In addition, Mfotie Njoya et al. (2017) [23] found the methanol extract from *S. pobeguinii* leaves and methanol extract from *S. pobeguinii* bark presented a high amount of alkaloids in comparison to the two other extracts (methanol extract from roots and CH_2_Cl_2_/MeOH extract from fruits) and this family of secondary metabolites can also be responsible for the cytotoxic effects [23]. In fact, several alkaloids isolated from plants are described as having anticancer effects. For example, the evidence based on in vivo and in vitro models indicated that isoquinoline alkaloids and/or isoquinoline-enriched plants exert significant anti-cancer effects through cell cycle arrest, apoptosis, and autophagy [49]. Finally, in the study performed by Mbaveng et al. (2019) [21] with *F. tessmannii*, among the different phytochemicals found, it appeared that only benphenanthridines presented cytotoxic effects. Regarding benzophenandrines, the presence of chloride appeared to significantly increase the cytotoxicity. Additionally, the cytotoxic effect may have been influenced by the presence of 8-OH (hydroxyl) and 7-OCH_3_ (methoxy) groups instead of two methoxy groups in C-8 and C-9, within the two chloride-containing benzophenanthridines [21].

In traditional medicine, plants are used as mixtures and not as single plants. Therefore, Ochwang’I et al. (2018) studied whether the combination of plant extracts would conduct to increased cytotoxicity against cancer cells. The authors selected aqueous extracts from Kenyan plants, which revealed poor cytotoxicity alone (*H. madagascariensis*, *Spathodea S. campanulate*, *P. africana*), and combined them with other extracts. The combined extracts exhibited stronger cytotoxic effects towards CRRF-CEM [34].

## 4. Conclusions

The results in this review give an important perspective for the anticancer activity research for African plant extracts. Some African plant extracts revealed a noteworthy anti-proliferative effect yielding lower IC_50_ values, e.g., 80% methanol extracts from *E. schimperiana* with IC_50_ values ranging from 1.85 ± 0.44 to 3.28 ± 1.2 µg/mL against A427, SiSo, and RT-4 cell lines; methanol extract from *S. pobeguinii* (leaves) with IC_50_ value of 10.19 µg/mL against HeLa cell line, methanol extract from *F. tessmannii* (bark) with an IC_50_ value of 17.34 µg/mL towards U87MG.ΔEGFR glioblastoma cells, or methanol extract of *S. vogelii* and *F. elastica* with IC_50_ values of 20 µg/mL. These values are under the range of 30 µg/mL that the NCI considers of interest for crude plant extracts. However, other extracts were found nontoxic at the studied concentrations on a panel of cancer cell lines (e.g., aqueous and ethanol extracts of the *K. africana* tuber). It is also worth mentioning that some extracts analyzed in these studies showed selective cytotoxicity on cancer cells and no significant toxicity to normal cells (e.g., *S. asperulus* dichloromethane extract, *A. laricinus* methanol extract, *S. pobeguinii*, *E. schimperiana*, *A. schimperi*, *K. foliosa*, and *K. petitiana* methanol extracts). In recent years, many phytochemicals isolated from African flora with potential anticancer effects have been identified. Therefore, an intensive screening must be pursued to increase cytotoxic molecules for further drug development. Furthermore, in the future, the use of plant extracts and/or their isolated compounds alone or in combination with conventional chemotherapy might be foreseen as interesting candidates as adjuvant drugs in cancer therapy. However, further studies are needed for the isolation of bioactive compounds from plant extracts and the assessment of the unknown effects as well as their synergistic effects in vitro and in an in vivo animal model. For most of the studies analyzed in this review, additional in vivo studies are needed to fully understand the impact of the reported results.

## Data Availability

Data sharing not applicable.

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
