# Peer review of "In Vitro Cytotoxic Activity of African Plants: A Review"

_molecules, 2022, doi:10.3390/molecules27154989_

Round 1

Reviewer 1 Report

In this review paper, Isabel Canga and others tried to provide cytotoxic potential and cell death mechanism of African plants through in vitro models like MCF-7, MDA-MBA-231, A549, and Caco-2 cells. In this paper, they analyzed 23 papers wherein 105 plants were selected for analysis using globally used research search platforms like Pubmed and Science direct. Information provided in the paper is not enough to get published in the current form and has to be revised majorly to improve the overall quality of this paper.

Some key comments are provided:

1.    The paper's abstract is poorly written and does not convey any special information. Initially, the authors talked about the cytotoxicity activity of African plants and then jumped to solvents used. It is making no sense. Therefore, some connection must be provided.

2.    The title suggests potential mechanisms, but unfortunately, there is not a single line related to the protective mechanisms of anti-cancer plants, so rewrite sections in the abstract and briefly describe the mechanism.

3.    The main body of the manuscript and title are not complementing each other, as the flow of the paper is not correct, authors are advised to make clear statements that will enhance the readability of the paper.

4.    Authors have mentioned IC50 values of some plants in the abstract. If authors are describing IC50, it should be clearly described every plant incorporated in the review otherwise this information is of no use.

5.    The starting of the introduction is not correct and the authors should have given it a better start with an explanatory paragraph about cancer which is missing.

6.    Authors haven’t incorporated anything about the mechanisms and pathways of carcinogenesis. The introduction fails to explain anything about ROS and other cancer information. Very less information in the introduction part.

7.    Page no 2 lines no 70 onwards is not understandable. It is not clear what the authors are trying to convey. Please check and correct accordingly.

8.    Why authors have given two separate headings to table 1?  All the information in the different tables should have been given in a single table with one column added as ‘native place of medicinal plants’ rather than giving separate tables.

9.    No information on nutraceuticals' protective mechanism or signaling pathways is given in the manuscript. Very general information is provided as the authors should have emphasized a specific part such as any protective pathway of plants.

10. Some ambiguity in the paper has to be corrected, since this is a review paper, what type of results do they want to describe? Please check it is correct.

11. The way of citation is not correct and there are few papers given that warrant a thorough literature survey (recent would be better).

12. Authors have made a lot of grammatical mistakes which can’t be neglected; therefore, significant English editing is required to provide clarity among readers.

13. A major revision is required.

Author Response

  1. The paper's abstract is poorly written and does not convey any special information. Initially, the authors talked about the cytotoxicity activity of African plants and then jumped to solvents used. It is making no sense. Therefore, some connection must be provided.

Some changes in the abstract were made in order to have the connection suggested by the reviewer. In our opinion now the abstract is in accordance with the objectives of the review.

  1. The title suggests potential mechanisms, but unfortunately, there is not a single line related to the protective mechanisms of anti-cancer plants, so rewrite sections in the abstract and briefly describe the mechanism.

The potential mechanisms of action of the studied plant extracts are explained in the discussion (for example: apoptosis and the induction of cell cycle arrest). These are the suggested mechanisms proposed in the papers included in this review. However, we rewrite the abstract and change the title because the focus of the review is the assessment of cytotoxic activity.

  1. The main body of the manuscript and title are not complementing each other, as the flow of the paper is not correct, authors are advised to make clear statements that will enhance the readability of the paper.

We changed the title and removed the “potential mechanisms of action”.

  1. Authors have mentioned IC50values of some plants in the abstract. If authors are describing IC50, it should be clearly described every plant incorporated in the review otherwise this information is of no use.

We removed the IC50 values from the abstract and left only a range and the limits suggested by the American National Cancer Institute USA (NCI), for a promising anticancer agent.

  1. The starting of the introduction is not correct, and the authors should have given it a better start with an explanatory paragraph about cancer which is missing.

We made some changes at the begging of the Introduction (with a new reference). For us, it´s not relevant to explore the carcinogenesis process, since it´s not the focus of this review. Similar papers only make a brief reference to cancer (focusing more on epidemiological data).

  1. Authors haven’t incorporated anything about the mechanisms and pathways of carcinogenesis. The introduction fails to explain anything about ROS and other cancer information. Very less information in the introduction part.

We complete the introduction with a short sentence referring the influence of ROS on tumor formation.

  1. Page no 2 lines no 70 onwards is not understandable. It is not clear what the authors are trying to convey. Please check and correct accordingly.

We removed those confusing lines.

  1. Why have authors given two separate headings to table 1?  All the information in the different tables should have been given in a single table with one column added as ‘native place of medicinal plants’ rather than giving separate tables.

We decided to join all the tables in one incorporating the Country column suggested by the referee. The review only has one single Table (Table 1).

  1. No information on nutraceuticals' protective mechanism or signaling pathways is given in the manuscript. Very general information is provided as the authors should have emphasized a specific part such as any protective pathway of plants.

The main subject of the review is only the cytotoxicity of the extracts, so the potential mechanisms of action referred in the discussion are the mechanism described in the analysed papers. However, we add a sentence related to plants protective mechanism or signaling pathways (a new reference was added).

  1. Some ambiguity in the paper has to be corrected, since this is a review paper, what type of results do they want to describe? Please check it is correct.

The results described are in accordance with the objectives of this review. To eliminate some ambiguity, we decided to add a general sentence in the end of the Introduction to focus on the objectives of this review.

  1. The way of citation is not correct and there are few papers given that warrant a thorough literature survey (recent would be better).

Citations have been reviewed and a few new references have been included

  1. Authors have made a lot of grammatical mistakes which can’t be neglected; therefore, significant English editing is required to provide clarity among readers.

A revision of the language was performed

  1. A major revision is required.

A revision of the language was performed.

Reviewer 2 Report

The manuscript entitled: "In vitro Cytotoxic Activity and Potential Mechanisms of Cell Death in African Plants: A Review" discusses the in vitro cytotoxic activity of the African plant crude extracts using cancerous cell lines. Authors have used PubMed and ScienceDirect databases for the literature review. Overall the work is sound and comprehensive. However, there are a few concerns. Please find my comments below.  

Overall, there are a few grammatical errors. 

Materials and methods are not required as a separate section. Authors should consider merging that into the introduction at an appropriate place. 

Line #20 "Different solvents..." should be removed. 

Line #22 - #25 is not clear. Rephrase the sentence/s. 

Reference #9 in Line #47: Adding the total number of cases will be helpful. Also, the reference is from 2012, I encourage authors to add more recent reference/s. 

Line #50 3.000 or 3,000?

Line #54 Reference needed. 

All the "in vitro" should be written as in vitro.

Tables are too wordy and the numbers are confusing. I strongly recommend shortening the text in results section. 

Also, Authors should consider reorganising the manuscript. One suggestion is dividing the table with texts. Each table should follow the respective text from discussion. That will get manuscript a better flow.  

Table #10: No information except plant names. 

Line #121 - #124: These lines should be shifted in the next section 3.1 cytotoxicity.

Line #148: Acetone extract? Is that correct?

Line #153: Curried out or Carried out? 

Author Response

The manuscript entitled: "In vitro Cytotoxic Activity and Potential Mechanisms of Cell Death in African Plants: A Review" discusses the in vitro cytotoxic activity of the African plant crude extracts using cancerous cell lines. Authors have used PubMed and ScienceDirect databases for the literature review. Overall, the work is sound and comprehensive. However, there are a few concerns. Please find my comments below.  

Thank you very much for your comment

Overall, there are a few grammatical errors. 

Materials and methods are not required as a separate section. Authors should consider merging that into the introduction at an appropriate place. 

This section was removed and incorporated (without a separate section) in the end of Introduction.

Line #20 "Different solvents..." should be removed. 

The sentence was removed.

Line #22 - #25 is not clear. Rephrase the sentence/s. 

The sentence was removed.

Reference #9 in Line #47: Adding the total number of cases will be helpful. Also, the reference is from 2012, I encourage authors to add more recent reference/s. 

We rephrase the sentence and add a more recent reference.

Line #50 3.000 or 3,000?

Indeed it was 3,000 and we modified. Thank you for notice it

Line #54 Reference needed.

The reference [9] was included

All the "in vitro" should be written as in vitro.

The change has been done along the manuscript.

Tables are too wordy and the numbers are confusing. I strongly recommend shortening the text in results section. Also, Authors should consider reorganising the manuscript. One suggestion is dividing the table with texts. Each table should follow the respective text from discussion. That will get manuscript a better flow.  

In view of the comment of both referees, we decided to join all the tables in one. Now, the review only has one general Table (Table 1). We also review the text along the table (because indeed, it was too wordy).

Table #10: No information except plant names. 

All the necessary information is now included in Table 1.

Line #121 - #124: These lines should be shifted in the next section 3.1 cytotoxicity.

We have moved the whole paragraph to the next section

Line #148: Acetone extract? Is that correct?

Yes, the reference [19] referred to acetone extract

Line #153: Curried out or Carried out? 

Thank you for noticing the mistake, it was carried out and it has been changed

Round 2

Reviewer 1 Report

Now paper is improved, however, English quality and references are not up to the mark and require serious consideration. I would strongly recommend suggest more related reference and English editing for language and grammatical error s for better readability. 

Author Response

Now paper is improved, however, English quality and references are not up to the mark and require serious consideration. I would strongly recommend suggest more related reference and English editing for language and grammatical error s for better readability. 

Answer:

Thank you very much for your comments that always contribute to improve our paper. We have checked all the references and we have changed some of them to more appropriate ones. Also, a revision of the language was performed, and several grammatical mistakes have been corrected.